# The Effect of C60 and Pentacene Adsorbates on the Electrical Properties of CVD Graphene on SiO_2_

**DOI:** 10.3390/nano13061134

**Published:** 2023-03-22

**Authors:** Jacopo Oswald, Davide Beretta, Michael Stiefel, Roman Furrer, Dominique Vuillaume, Michel Calame

**Affiliations:** 1Empa, Swiss Federal Laboratories for Materials Science and Technology, Transport at Nanoscale Interfaces Laboratory, Überlandstrasse 129, CH-8600 Dübendorf, Switzerland; 2Swiss Nanoscience Institute, University of Basel, Klingelbergstrasse 82, CH-4056 Basel, Switzerland; 3Centre National de la Recherche Scientifique, Institute for Electronic, Microelectronic and Nanotechnology (IEMN), 59652 Villeneuve d’Ascq, France; 4Department of Physics, University of Basel, Klingelbergstrasse 82, CH-4056 Basel, Switzerland

**Keywords:** organic, semiconductor, graphene, field effect, transistor, van der Waals, C60, Pentacene, hybrid, heterostructures

## Abstract

Graphene is an excellent 2D material for vertical organic transistors electrodes due to its weak electrostatic screening and field-tunable work function, in addition to its high conductivity, flexibility and optical transparency. Nevertheless, the interaction between graphene and other carbon-based materials, including small organic molecules, can affect the graphene electrical properties and therefore, the device performances. This work investigates the effects of thermally evaporated C60 (*n*-type) and Pentacene (*p*-type) thin films on the in-plane charge transport properties of large area CVD graphene under vacuum. This study was performed on a population of 300 graphene field effect transistors. The output characteristic of the transistors revealed that a C60 thin film adsorbate increased the graphene hole density by (1.65 ± 0.36) × 10^12^ cm^−2^, whereas a Pentacene thin film increased the graphene electron density by (0.55 ± 0.54) × 10^12^ cm^−2^. Hence, C60 induced a graphene Fermi energy downshift of about 100 meV, while Pentacene induced a Fermi energy upshift of about 120 meV. In both cases, the increase in charge carriers was accompanied by a reduced charge mobility, which resulted in a larger graphene sheet resistance of about 3 kΩ at the Dirac point. Interestingly, the contact resistance, which varied in the range 200 Ω–1 kΩ, was not significantly affected by the deposition of the organic molecules.

## 1. Introduction

Hybrid van der Waals heterostructures made of Graphene (Gr) and Organic Semiconductors (OSC) are being widely investigated for their potential applications in sensors [1,2], solar cells [3], light emitting diodes [4] and vertical transistors [5,6,7,8,9,10,11,12,13]. The latter, which combine very short vertical OSC channels with a graphene electrode, exploit the weak electrostatic screening of graphene and the field-tunable charge injection barrier at the Gr/OSC interface to control the current in the vertical channels. In this context, understanding the effects of *n*- and *p*-type organic molecules at the Gr/OSC interfaces is crucial to enable the realization of efficient graphene-based complementary circuits, the operating mechanisms of which will ultimately be limited by the graphene conductivity and contact resistance. In fact, the van der Waals interactions of carbon materials, including small organic molecules, e.g., C60-Fullerene and Pentacene, have an important influence on the electrical properties of graphene and therefore, on the device performances. While the growth mechanisms and orientation of C60 and Pentacene molecules on graphene have been widely studied [7,14,15,16,17], a deeper understanding of the Gr/Organic Semiconductor (OSC) interface formation and of the charge transport in graphene is still necessary to achieve desirable electronic device properties and functionalities.

This work investigates the effect of C60 and Pentacene adsorbates on the electrical properties of large-area graphene grown by Chemical Vapor Deposition (CVD). C60 and Pentacene molecules are commonly used as *n*- and *p*-type organic semiconductors [18,19,20] that can be thermally evaporated on the target substrate, and are therefore of interest because they can be used for graphene-based complementary circuits. Thin films of C60 and Pentacene were thermally evaporated on two distinct sets of Graphene Field Effect Transistors (GFETs). The surface morphology and chemical composition of the hybrid Gr/C60 and Gr/Pentacene heterostructures, which formed the GFET channels, were investigated using Atomic Force Microscopy (AFM) and Raman spectroscopy. The residual doping and charge carrier mobility in graphene were obtained from the output characteristics of the GFETs in vacuum. Then, the Fermi energy shift induced by the C60 and Pentacene was estimated using the linear energy dispersion relation of graphene. The Transfer Length Method (TLM) was used to extrapolate the sheet and contact resistance of graphene. Finally, the electrical properties of graphene before and after the deposition of the C60 and Pentacene molecules were compared and summarized.

## 2. Results and Discussion

This study was performed on two sets of 150 Graphene Field Effect Transistors (GFETs) with a fixed channel width *W* of 5 μm and five different channel lengths *L* of 5, 10, 20, 50 and 100 μm. The two sets include 30 GFETs per channel length and were fabricated on two separate chips (Appendix A). The detailed fabrication protocol of the two chips and the Chemical Vapor Deposition (CVD) process used to grow the graphene sheets are reported in the Experimental Methods section. Briefly, the Cu foil that was used as substrate for the CVD growth of graphene was etched away using Ferric Chloride (FeCl_3_). The remaining graphene sheet was transferred on the pre-patterned Ti/Au electrodes (5 nm/50 nm) on a Si/SiO_2_ (525 μm/300 nm) substrate and patterned into geometrically defined channels by e-beam lithography. Then, the molecules were thermally evaporated, without using any physical mask nor photolithographic patterning, resulting in a uniform coverage of the whole chip. Figure 1a shows the optical microscope image of a representative GFET with *L* = 50 μm and *W* = 5 μm. The output and transfer characteristics of the GFETs were measured in vacuum before and after depositing the molecules, after 24 h of vacuum exposure to assure complete desorption of the moisture and water. Figure 1b shows the electric circuit and cross-section schematics of a GFET coated with C60 or Pentacene. Figure 1c shows the Highest Occupied Molecular Orbital (HOMO) and Lowest Unoccupied Molecular Orbital (LUMO) energy levels of the two Organic Semiconductors (OSCs) and the Fermi energy of graphene, i.e., −4.6 eV, with respect to the vacuum level, according to the literature [18]. The reported values of the HOMO and LUMO energy levels of C60 were about −6.4 eV and −4.1 eV [18,21], while for Pentacene, they were about −5.1 eV and −2.9 eV, respectively [18,22,23]. The HOMO and LUMO energy levels of the OSCs were extrapolated from Ultraviolet Photoelectron Spectroscopy (UPS) and Inverted Photoelectron Spectroscopy (IPES) measurements, and defined at the onset of the Gaussian Density of States (DOS) [22], as roughly shown in Figure 1c. The Fermi energy of graphene lies within about 0.5 eV from the HOMO energy level of Pentacene and within about 0.5 eV from the LUMO energy level of C60. Accordingly, one would expect that C60 and Pentacene adsorbates act like electron acceptors and donors, respectively. However, the energetic of the interfaces is not trivial, and many factors can play a role, including (i) intragap defects, which could result in occupied and unoccupied states, (ii) dipole formation, (iii) Fermi-level pinning and/or (iv) the presence of impurities, e.g., H_2_O and/or resist residues [24,25,26,27,28,29]. Therefore, the effect of the adsorbates on the graphene properties cannot be forecasted by simple arguments on the energy diagrams of the isolated systems. On the other hand, one can estimate the residual charge density from the transfer characteristics of the GFETs in a controlled environment. In addition, a combination of Raman spectroscopy and Atomic Force Microscopy (AFM) gives important insight on the morphology and chemical composition of the Gr/OSC hybrid heterostructure.

Figure 2a,b shows the AFM images and profiles of two representative GFET channels coated with thin films of C60 (Chip 1) and Pentacene (Chip 2), respectively. The average measured thickness of the C60 and Pentacene layers on the SiO_2_ substrates were about 10 nm and 5 nm, as shown in the Appendix A. The height difference and morphology were attributed to the different growing mechanisms of C60 and Pentacene on SiO_2_ and graphene. The AFM profile of Figure 2c reveals a uniform C60 thin film growth on both SiO_2_ and graphene, as was previously reported [7]. The average height step of about 1 nm (red line) on the GFET channel was due to graphene, and it was in the range of previously reported values for pristine graphene, i.e., 0.4–1.7 nm [30]. The discrepancy between the measured and expected step thickness of graphene (about 0.3 nm) was therefore not surprising and could be attributed to different components, including the graphene-adsorbate layer, tip-surface interactions and imaging force [30]. Here, the very thin layer of C60 allowed for the observation of the typical features of the underlying CVD graphene sheet (e.g., wrinkles caused by the Cu catalyst grain boundaries [31]). The AFM image of Figure 2b shows that thermally evaporated Pentacene grew in the typical thin film phase on SiO_2_ [32], whereas it grew in 3D elongated islands on graphene [14,16]. Figure 2d shows that the average channel step thickness was approximately 30 nm (red line), and that the Pentacene film on graphene formed an irregular profile (black line) of 30–50 nanometer-thick elongated islands. The thick Pentacene islands masked the typical features of the underlying CVD graphene, which were not visible in the AFM image. Nevertheless, the graphene channel was clearly distinguishable due to the different growth mechanisms of Pentacene on SiO_2_ and on the graphene channel. The presence of the graphene channel under the organic thin films was further confirmed using Raman spectroscopy. Figure 2e,f shows the average Raman spectra obtained from each set of 150 devices before and after deposition of the thin films. Refer to the Experimental Methods section and the Appendix A for the details on the Raman signal acquisition, averaging, background subtraction and normalization. The Raman spectra of wet transferred CVD graphene (Gr on Si/SiO_2_ substrate) showed the characteristic G (1587 cm^−1^) and 2D (2681 cm^−1^) peaks, as well as the D (1340 cm^−1^) peak [33], with a weak amplitude, possibly resulting from fabrication-induced defects. The typical first-order optical mode (520 cm^−1^) and second-order scattering band (940–980 cm^−1^) of the Si substrate [34] were observed in all Raman spectra. Figure 2e shows the chemical composition of the GFETs (Chip 1) before (Gr) and after (Gr/C60) the deposition of C60. The hybrid Gr/C60 layer displayed the characteristic Raman active vibrations of C60 [35], i.e., H_1g_ at 264 cm^−1^, H_2g_ at 430 cm^−1^, A_1g_ at 491 cm^−1^, H_3g_ at 707 cm^−1^, H_4g_ at 772 cm^−1^, H_5g_ at 1099 cm^−1^, H_6g_ at 1242 cm^−1^, A_2g_ at 1462 cm^−1^ and H_8g_ at 1568 cm^−1^, as well as the characteristic Raman peaks of CVD graphene (G at 1587 cm^−1^ and 2D at 2681 cm^−1^) observed before the deposition of C60. Figure 2f shows the chemical composition of the GFETs (Chip 2) before (Gr) and after (Gr/Pentacene) the deposition of Pentacene. The hybrid Gr/Pentacene layer showed the characteristic Raman features of Pentacene [36]. The peaks at 1158, 1177, 1371, 1410, 1456, 1499 and 1533 cm^−1^ were assigned to the A_g_ bands, while the peak at 1595 cm^−1^ was assigned to the B_3g_ band. The 2D peak (2687 cm^−1^) of the CVD graphene could be clearly distinguished, whereas the G peak overlapped with the B_3g_ band of Pentacene. The AFM surface morphology combined with Raman chemical analysis confirmed the presence of the geometrically well-defined Gr/C60 and Gr/Pentacene hybrid heterostructures.

Figure 3a–d shows the average transfer characteristics (*I_DS_* vs. *V_GS_*) of pristine GFETs and coated GFETs measured under vacuum and at room temperature. The standard deviations are indicated by the shaded areas. Refer to the Experimental Methods section for details on the data filtering and population selection process and to the Appendix A for all the individual output characteristics (Appendix A), transfer characteristics (Appendix A) and information on the population (Appendix A). All the devices showed the typical output characteristic (*I_DS_* vs. *V_DS_*) and transfer characteristic (*I_DS_* vs. *V_GS_*) of graphene field effect transistors [37]. The transfer characteristics displayed negligible hysteresis (Appendix A), and thus only the continuous backward traces were considered. All C60-GFETs (Chip 1) showed $I_{GS}\ll I_{DS}$ on the entire *V_GS_* range (Appendix A), while all Pentacene-GFETs (Chip 2) displayed a pronounced gate-leakage at high negative gate voltages, a phenomenon that does not affect the conclusion of this work. As expected, the *I_DS_* was inversely proportional to the graphene channel length L ($I_{DS}\sim 1/L$) and the position of the Dirac point $V_{GS}^{Dirac}$, which corresponds to the minimum of the *I_DS_*, was found at positive and close to zero *V_GS_* for the two sets of pristine GFETs, i.e., $V_{GS}^{Dirac}=2.8\pm 9.1\;V$ for Chip 1 and $V_{GS}^{Dirac}=1.3\pm 6.7\;V$ for Chip 2. This indicates that the fabrication process resulted in clean graphene, possibly with few PMMA resist residues and thus a relatively low p-doping level [24,25,26,27]. Figure 3b,d shows that the distribution of the Dirac point of the GFETs was shifted to $V_{GS}^{Dirac}=23.0\pm 5.0\;V$ after the deposition of C60 (Chip 1), while it was shifted to $V_{GS}^{Dirac}=-7.6\pm 7.5\;V$ after the evaporation of Pentacene (Chip 2). Assuming that the Dirac point shift was solely due to the charge transfer between graphene and the molecules, it was possible to estimate the residual doping in graphene. In fact, the relation between the charge carrier density *n* in the graphene channel and *V_GS_*, neglecting the quantum capacitance [37,38], is given by the electrostatic capacitance. In Formula [27],
(1)$$n=\frac{C_{GS}}{e}\left|V_{GS}-V_{GS}^{Dirac}\right|$$
where e is the elementary charge and $C_{GS}=\epsilon_{0}\epsilon_{r}/t$ is the gate capacitance per unit area, where $\epsilon_{0}$ is the vacuum permittivity, while $\epsilon_{r}=3.9$ [39] and t=300 nm are the dielectric constant and thickness of SiO_2_, respectively. Therefore, the residual doping in graphene due to the molecules is given by $n_{0}=n(V_{GS}=0)$. When $V_{GS}>V_{GS}^{Dirac}$, electrons are accumulated in the graphene channel, while for $V_{GS}<V_{GS}^{Dirac}$, holes are accumulated in the graphene channel. The deposition of C60 molecules led to a graphene hole density of *n*_0_ = (1.65 ± 0.36) × 10^12^ cm^−2^, against the residual doping of pristine graphene with *n*_0_ = (0.20 ± 0.65) × 10^12^ cm^−2^ (Chip 1). For the Pentacene molecules, the graphene was electron-doped with an electron density of *n*_0_ = (0.55 ± 0.54) × 10^12^ cm^−2^, against the residual hole doping of graphene before the deposition with *n*_0_ = (0.09 ± 0.48) × 10^12^ cm^−2^ (Chip 2). Such a charge transfer could not be anticipated a priori from the HOMO and LUMO energy levels of the individual molecules relative to the graphene Fermi level only, as illustrated for C60 and Pentacene in Figure 1c. The corresponding graphene Fermi level shift at $V_{GS}=0V$ could be extrapolated from the linear energy dispersion relation of graphene [40], i.e., $\Delta E={E_{DP}-E}_{F}=-sgn\left(V_{GS}^{Dirac}\right)\hbar v_{F}\sqrt{\pi n_{0}}$, where $E_{DP}$ is the energy of the Dirac point, $E_{F}=-4.6eV$ is the Fermi energy and $v_{F}=10^{6}\;m/s$ is the Fermi velocity in graphene. The deposition of C60 on graphene resulted in a Fermi energy downshift of ΔE=−150±15 meV, against ΔE=−52±55 meV for the pristine graphene (Chip 1). The deposition of Pentacene on graphene resulted in a Fermi energy upshift of ΔE=+86±35 meV, against ΔE=−36±53 meV for pristine graphene (Chip 2). The charge carrier density $n_{0}$ and Fermi energy shift ΔE induced by C60 on graphene were similar to previously reported values for Gr/C60 hybrid layers obtained from Raman and THz-TDS measurements [41]. The field effect transistor measurements, compared to other techniques, allow to also determine the hole mobility $\mu_{h}$ and electron mobility $\mu_{e}$ from the transfer characteristic. In fact, the field effect electron and hole mobility reads [37],
(2)$$\mu =\frac{Lg_{m}}{WC_{GS}V_{DS}}$$
where $g_{m}=max(dI_{DS}/dV_{GS})$ is the maximum transconductance at $V_{DS}=50\;mV$ [42]. The hole mobility and electron mobility were obtained from the transfer characteristics for $V_{GS}<V_{GS}^{Dirac}$ and for $V_{GS}>V_{GS}^{Dirac}$, respectively. Refer to the Experimental methods and Appendix A for the details on the numerical derivative of the *I_DS_* vs. *V_GS_* traces. Figure 3e–j shows the distributions of the Dirac point $V_{GS}^{Dirac}$, the electron mobility $\mu_{e}$ and hole mobility $\mu_{h}$ of the GFETs before and after the deposition of the molecules. In the C60-GFETs measurements (Chip 1), the increase in the residual hole density *n*_0_ was accompanied by a decrease, from $\mu_{e}=1430\pm 354\;{cm}^{2}V^{-1}s^{-1}$ to $\mu_{e}=713\pm 261\;{cm}^{2}V^{-1}s^{-1}$, in the electron mobility, possibly due to an increased charge-impurity scattering [43]. The hole mobility remained almost unaffected, as shown in Figure 3f,g. Similarly, the deposition of Pentacene on the GFETs (Chip 2) led to an increase in the residual electron density *n*_0_ which was accompanied by a decrease, from $\mu_{h}=1910\pm 310\;{cm}^{2}V^{-1}s^{-1}$ to $\mu_{h}=1268\pm 275\;{cm}^{2}V^{-1}s^{-1}$, in the hole mobility. The electron mobility was also slightly reduced, as shown in Figure 3i,j. It is worth mentioning that the mobility estimated according to Equation (2) was a lower bound to the actual GFET mobility, as the inflection point of the transfer characteristic might have occurred beyond the chosen *V_GS_* range.

Figure 4a,b shows the GFETs’ sheet resistance ($R_{S}$) and contact resistance ($R_{C}$) extrapolated from the transfer characteristic of Figure 3a–d using the Transfer Length Method (TLM), where the total resistance is $R_{T}=2R_{C}+R_{S}$, as described in the Experimental Methods and in the Appendix A. The negative average $R_{C}$ values from the uncertainty of the linear regression were discarded (hatched area). On the one hand, the sheet and contact resistances of the two sets of pristine GFETs (Figure 4a,b) showed similar characteristics: (i) they both depended on $V_{GS}$; (ii) the maximum of the sheet resistances, which was approximately 3 kΩ, was found in proximity of the Dirac point $V_{GS}^{Dirac}$; and (iii) the maximum of $R_{C}$, approximately 1 kΩ, was found for $V_{GS}<V_{GS}^{Dirac}$. The latter suggested that the hole and electron energy barrier heights at the Au/Gr interface, which result from the charge transfer between the metal electrodes and graphene [44,45], were different [44]. On the other hand, after the deposition of the molecules: (i) the electrostatic doping reduced the sheet resistance of graphene, which approached approximately 1 kΩ at $V_{GS}=\pm 50\;V$; (ii) the maximum of *R_S_* increased by about 200 Ω and shifted to a more positive $V_{GS}$ for the C60-GFETs and to a more negative $V_{GS}$ for the Pentacene-GFETs, which was possibly due to an increased charge-impurity scattering resulting in a reduced charge carrier mobility [43], assuming that the graphene charge carrier density at $V_{GS}^{Dirac}$ was not affected by the molecules; (iii) the maximum of *R_C_* shifted with *R_S_* and remained close to the Dirac point, suggesting that the charge carrier injection mechanism at the Au/Gr interface was not significantly affected by the molecules. Assuming that the charge injection mostly occurred at the edge between the metal electrodes and Gr (*W* = 5 μm), the gate dependent contact resistivity $\rho_{C}=R_{C}W$ could be calculated and spanned the range of 0.5–5 kΩ μm. Table 1 summarizes all the electrical properties of the GFETs. It is worth observing, that taking into account the graphene Fermi energy shift induced by the OSCs, and using the reported HOMO and LUMO energy levels of C60 and Pentacene [18], the nominal energy barriers at $V_{GS}=0\;V$ for electrons ($\Phi_{B0,e}=E_{HOMO}-E_{F,Gr}$) at the Gr/C60 and for holes ($\Phi_{B0,h}=E_{F,Gr}-E_{LUMO}$) at the Gr/Pentacene interfaces, resulted in about 0.6 eV, which is within ±0.1 eV of the previously reported values obtained from the thermionic emission model [7,13]. This supports the hypothesis that all interfacial phenomena at the C60/graphene and Pentacene/graphene interfaces account for only 15% of the nominal energy barriers that can be estimated from the energy diagram of the isolated molecules. These Schottky-like interfaces are favorable for the realization of graphene-based vertical organic transistors that exploit complementary *n*- and *p*-type organic semiconductors [7,8,9,10,11,12].

## 3. Conclusions

In summary, CVD graphene field effect transistors (GFETs) with a fixed channel width (5 µm) and different channel lengths (5, 10, 20, 50 and 100 µm) were fabricated and electrically characterized under vacuum. Subsequently, thin films of C60 and Pentacene were deposited on two distinct GFET chips by thermal evaporation and electrically characterized again under vacuum. The GFETs’ transfer characteristic revealed that the deposition of C60 on graphene results in an increased residual graphene hole density of (1.65 ± 0.36) × 10^12^ cm^−2^, whereas the deposition of Pentacene results in an increased residual graphene electron density of (0.55 ± 0.54) × 10^12^ cm^−2^. In both cases, the increase in the residual charge carriers was accompanied by a reduced charge mobility, possibly due to an increased charge-impurity scattering in graphene. The Fermi energy of graphene shifted to (−150 ± 15) meV after the deposition of C60, and it shifted to (+86 ± 35) meV after the deposition of Pentacene, while the charge carrier injection at the Au/Gr did not seem to be significantly affected. Overall, this work provides useful insight into the graphene in-plane charge transport and on the energetic of the Gr/C60 and Gr/Pentacene hybrid heterostructures, which could be exploited in more complex organic electronic devices. For instance, the energy barriers forming at the Gr/C60 and Gr/Pentacene interfaces and the weak electrostatic screening of graphene could be beneficial for forming *n*- and *p*-type channels in graphene-based vertical organic transistors.

## 4. Experimental Methods

### 4.1. Chemical Vapor Deposition (CVD) of Graphene

Graphene was grown in-house by chemical vapor deposition (CVD) on copper foils (Thermoscientific, Alfa Aesar, Kandel, DE, 25 μm thick, annealed, uncoated, 99.8%) with a fully automated setup. The foils were ultrasonicated in Acetone, rinsed with Isopropyl Alcohol (IPA) and dried with N_2_. Then, the foils were placed in Acetic Acid (CH_3_COOH) for 30 min, rinsed in de-ionized water (DIW) and Ethanol and dried with N_2_. The cleaned foils were pre-annealed at 1000 °C for 1 h in a H_2_ (20 sccm) and Ar (200 sccm) atmosphere in the CVD oven (approximately 1 mbar). Then, graphene grew for 35 min in a CH_4_ (0.05 sccm), H_2_ (20 sccm) and Ar (200 sccm) atmosphere under a pressure of about 120 mbar. After the CH_4_ flow was stopped, the CVD oven was left to cool down to room temperature (at approximately 1 mbar).

### 4.2. Wet Transfer Method of CVD Graphene

The CVD graphene was transferred as previously reported [42,46]. To protect graphene, a film of PMMA (50k) was spin-coated on the top side of the CVD graphene/Cu foils. Then, the graphene grown on the Cu back side was removed using Reactive Ion Etching (RIE) in an Ar (15 sccm) and O_2_ flow (30 sccm) for 2 min. The Cu substrate was etched away in a Ferric Chloride (FeCl_3_, Transene CE-100) bath for 1 h, and then transferred to de-ionized water (DIW). Subsequently, the remaining graphene/PMMA sheet was placed in hydrochloric acid (HCl 10%) for 5 min and transferred back to DIW. Finally, graphene could be transferred onto the target substrate and dried overnight in vacuum oven (approximately 1 mbar, 80 °C).

### 4.3. Graphene Field Effect Transistors (GFETs) Fabrication

Two chips including 150 Graphene Field Effect Transistors (GFETs) were fabricated. Graphene sheets grown by CVD on Cu foils were transferred onto photolithography pre-patterned Ti (5 nm)/Au (50 nm) electrodes on Si (525 μm)/SiO_2_ (300 nm) substrates. The Au bottom contacts architecture (Figure 1b) was chosen to minimize the number of lithography steps after the transfer of graphene. A double layer of PMMA 50K/950K (AR-P 632-06/AR-P 672.02) was spin-coated on the graphene, exposed to an e-beam and developed for 1 min in MIBK:IPA (1:2). Then, the samples were rinsed in IPA and dried with N_2_. Finally, the unprotected graphene was etched away using RIE for 30 s (15 sccm Ar, 30 sccm O_2_). The remaining PMMA, protecting the graphene channels, was removed in Acetone (55 °C) for 1 h and IPA (55 °C) for 1 h.

### 4.4. Thermal Evaporation of C60 and Pentacene

Fullerene-C60 (99.9%, sublimed) powder was purchased from Sigma-Aldrich (Buchs, CH), while Pentacene powder (99.999%, purified by sublimation) was purchased from Tokyo Chemical Industry (TCI, Eschborn, DE). The two types of molecules were thermally evaporated under vacuum (approximately 10^−6^ mbar) without further treatments. The C60 thin film was deposited by thermal evaporation at about 0.2 Å/s from a molybdenum evaporation boat (Umicore). The Pentacene thin film was thermally evaporated at about 0.05 Å/s using a low-temperature-controlled source (about 120 °C) with an Al_2_O_3_ crucible (Creaphys). The evaporation rate was maintained by monitoring the quartz crystal microbalance of the evaporator. The temperature of the target substrate was not actively controlled during the evaporation.

### 4.5. Electrical Characterization

The devices were electrically characterized under vacuum (approximately 10^−6^ mbar) and at room temperature. The samples were shortly exposed to ambient conditions during the transport from the evaporation chamber to the electrical prober vacuum chamber. They were kept in vacuum conditions (approximately 10^−6^ mbar) for at least 24 h before being characterized. The electronics comprised an AdWin Gold II ADC-DAC unit, a low-noise current-to-voltage converter (Femto DDPCA-300, FEMTO Messtechnik, Berlin, DE) to measure the drain-to-source current (*I_DS_*), and a high-voltage amplifier (Basel SP908, University of Basel, Electronic Lab, Basel, CH) to provide the gate-to-source voltage (*V_GS_*). The drain-to-source voltage (*V_DS_*) was directly provided by the ADC-DAC. The automated probe station and the ADC-DAC were controlled via LabVIEW and Matlab scripts. In the drain-to-source voltage (*V_DS_*) sweep from −50 mV to +50 mV, the voltage step was set to 0.5 mV, the sweep rate to 50 mV/s and the gate-to-source voltage was set to 0 V. In the gate-to-source voltage (*V_GS_*) sweep from −50 V to +50 V, the voltage step was set to 0.1 V, the sweep rate to 10 V/s and the drain-to-source voltage (*V_DS_*) was set to 50 mV. The internal averaging was set to 20 ms for all the measurements. Only the backward sweep traces were considered for the analysis, refer to the Appendix A for the full sweeps (Appendix A). The gate-to-source current (Appendix A) was measured using a Semiconductor Parameter Analyzer Keithley 4200 (Keithley, Tektronix, Koln, DE).

### 4.6. Atomic Force Microscopy (AFM)

The height images were measured under ambient conditions using a Bruker Icon AFM (Bruker Corporation, Billerica, MA, USA) in tapping mode. The AFM was equipped with a TESPA-V2 cantilever with a tip apex radius of 7 nm, with a resonant frequency of 320 kHz and a spring constant of 37 N/m. The AFM data were processed with Gwyddion (version 2.56) which was used to extract the single and average height profiles and export the images.

### 4.7. Raman Spectroscopy

Raman spectra were acquired in ambient conditions using a 532 nm excitation wavelength with a WITec Alpha 300R confocal Raman microscope mounting an 100× objective (EC Epiplan-Neofluar Dic 100×/0.90, Zeiss) and a 300 mm lens-based spectrometer (grating: 600 g mm^−1^) equipped with a TE-cooled charge-coupled device (Andor Newton, Oxford Instruments). One Raman spectrum for each device was collected (150 devices per chip). The laser power was set to 0.1 mW and 0.5 mW for the Gr/C60 and Gr/Pentacene heterostructures, respectively. In both cases, the integration time was set to 30 s. The average Raman spectra of graphene, Gr/C60 and Gr/Pentacene were obtained using all the Raman spectra acquired on each set of devices. Refer to Appendix A for details on the Raman spectra processing.

### 4.8. Data Selection and Analysis

Python scripts were used for the data selection, analysis and visualization. Raman spectra averaging and polynomial background subtraction was done using numpy [47]. The results of the electrical measurements were organized in pandas [48] dataframes. The devices showing a non-linear output characteristic, low $I_{DS}\ll 1nA$ (open circuit) and/or more than one minimum in the transfer characteristic were discarded from the analysis. Only the backward sweep of the transfer characteristics was considered for the determination of the Dirac point, the mobility and the resistances. The GFETs electron and hole mobility were extrapolated using Equation (2) and the maxima of the transconductance in the $V_{GS}$ range –50 to +50 V. A Savitzky–Golay filter with window size 20 and smoothing order 3 was applied to the transfer characteristics before computing the numerical derivative $dI_{DS}/dV_{GS}$. A Transfer Length Method (TLM), i.e., $R_{T}=2R_{C}+R_{S}$, was implemented using scipy [49]. The $R_{C}$ and $R_{S}$ were obtained from the intercept and slope of the weighted linear regression of $R_{T}$ vs. *L*. The weights were set to ${1/R}_{T}$. Refer to the Appendix A for details on the Raman spectra processing (Appendix A), the numerical derivative (Appendix A) of the drain-to-source current $dI_{DS}/dV_{GS}$ and for the gate voltage dependent TLM (Appendix A).

## Figures and Tables

**Figure 1 nanomaterials-13-01134-f001:**
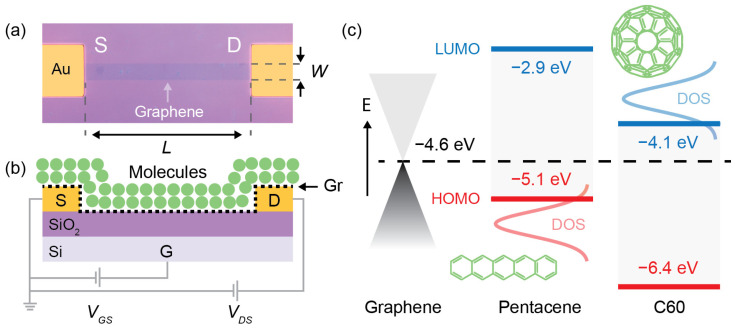
(**a**) Optical microscope image of a representative Graphene Field Effect Transistor (GFET). The channel length *L* and width *W* of this device are 50 μm and 5 μm, respectively. The image shows the source (S) and drain (D) gold electrodes on the Si/SiO_2_ substrate. (**b**) Electrical schematic of the GFET (not to scale). The cross-section shows the heavily p-doped Si Gate (G), the SiO_2_ dielectric (300 nm), the Ti/Au source/drain (5 nm/50 nm) and the Gr channel coated with C60 or Pentacene molecules. The source (S) electrode is connected to ground. In the gate-to-source voltage (*V_GS_*) sweep, the drain-to-source bias (*V_DS_*) is kept constant while the current (*I_DS_*) is measured. (**c**) Energetic representation of the Highest Occupied Molecular Orbital (HOMO) and Lowest Unoccupied Molecular Orbital (LUMO) nominal levels of the C60 and Pentacene molecules with respect to the Fermi energy level of pristine graphene (black dashed line). The Fermi energy of graphene lies within 0.5 eV from the HOMO energy level of Pentacene and within 0.5 eV from LUMO energy level of C60.

**Figure 2 nanomaterials-13-01134-f002:**
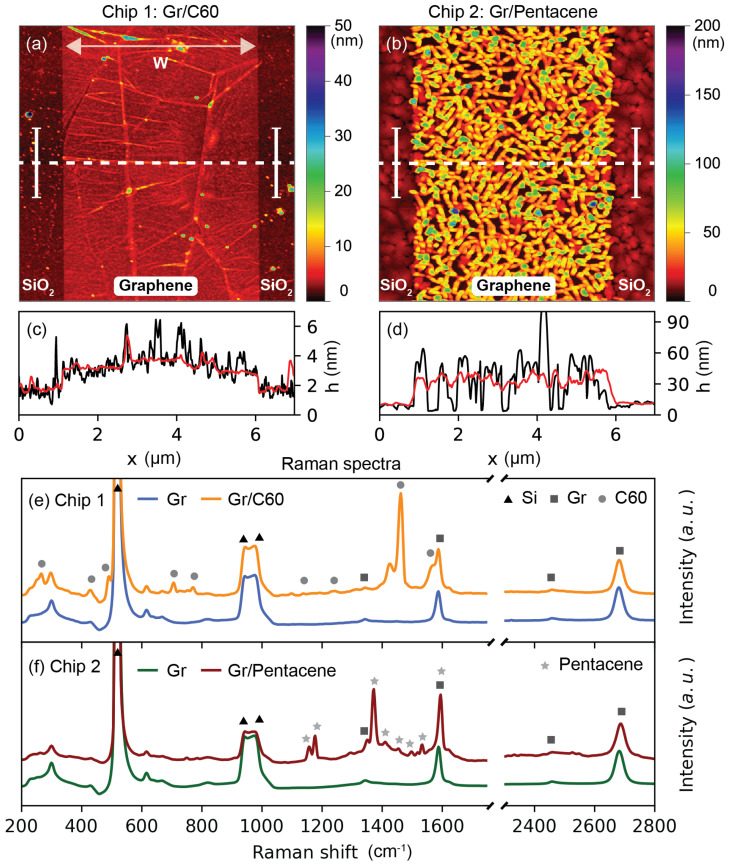
Surface morphology and chemical composition of the Gr/C60 (Chip 1) and Gr/Pentacene (Chip 2) heterostructures. (**a**) AFM height image of a representative Gr/C60 FET channel. (**b**) AFM height image of a representative Gr/Pentacene FET channel. (**c**) AFM height profile (black line) of the Gr/C60 FET channel taken along the dashed white line in (**a**). The red line is the mean value of 128 height profiles shown by the white bars in the AFM image. (**d**) AFM height profile (black line) of the Gr/Pentacene FET channel taken along the dashed white line in (**b**). The red line is the mean value of 128 height profiles shown by the white bars in the AFM image. (**e**) Average Raman spectra of the C60-GFETs (Chip 1). The Raman spectrum of the pristine CVD graphene is blue, while the Raman spectrum of Gr/C60 is orange. (**f**) Average Raman spectra of the Pentacene-GFETs (Chip 2). The Raman spectrum of the pristine CVD graphene is green, while the Raman spectrum of Gr/Pentacene is red. All spectra are normalized to the 2D peak of graphene. The triangle, square, circle and star symbols represent the characteristic Raman peaks of Si [34], Gr [33], C60 [35] and Pentacene [32], respectively.

**Figure 3 nanomaterials-13-01134-f003:**
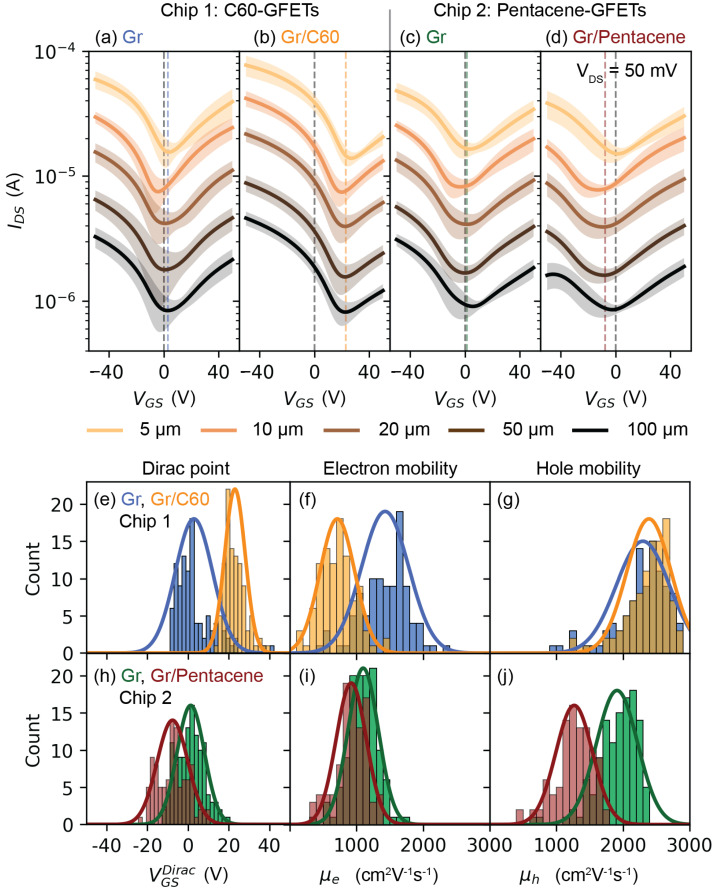
GFETs’ electrical measurements. (**a**) Average transfer characteristic (*I_DS_* vs. *V_GS_*) of pristine CVD graphene FETs before C60 deposition (Chip 1, 101 devices, all *L* included). The shaded areas are the standard deviations. (**b**) Average transfer characteristic of the C60-GFETs (Chip 1, 101 devices, all *L* included). (**c**) Average transfer characteristic of pristine CVD graphene before Pentacene deposition (Chip 2, 119 devices, all *L* included). (**d**) Average transfer characteristic of the Pentacene-GFETs (Chip 2, 98 devices, all *L* included). (**e**) Histograms of the graphene Dirac position ($V_{GS}^{Dirac}$) before (blue) and after deposition of C60 (orange). (**f**) Histograms of the graphene electron mobility ($\mu_{e}$) before (blue) and after deposition of C60 (orange), all *L* included. (**g**) Histograms of the graphene hole mobility ($\mu_{h}$) before (blue) and after deposition of C60 (orange), all *L* included. (**h**) Histograms of the graphene Dirac position ($V_{GS}^{Dirac}$) before (green) and after deposition of Pentacene (red), all *L* included. (**i**) Histograms of the graphene electron mobility ($\mu_{e}$) before (green) and after deposition of Pentacene (red), all *L* included. (**j**) Histograms of the graphene hole mobility ($\mu_{h}$) before (green) and after deposition of Pentacene (red), all *L* included. Bins width of graphene Dirac point position and mobility histograms are 2 V and 100 cm^2^V^−1^s^−1^, respectively.

**Figure 4 nanomaterials-13-01134-f004:**
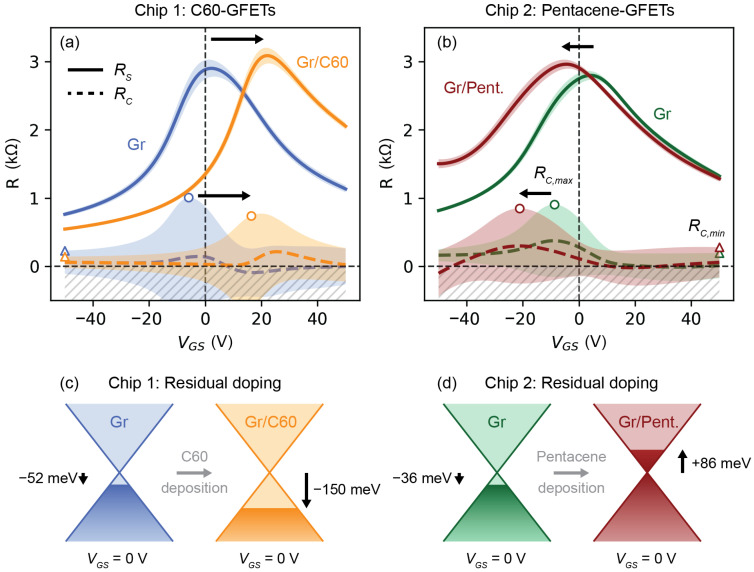
GFETs’ sheet resistance ($R_{S}$) and contact resistance ($R_{C}$) represented by solid and dashed lines, respectively. (**a**) $R_{S}$ (solid line) and $R_{C}$ (dashed line) of pristine CVD graphene before (blue) and after (orange) deposition of C60 (Chip 1). The circles and triangles show the maxima and minima of the contact resistance, respectively. The arrows show the shift of the maximum of $R_{S}$ and of the maximum of $R_{C}$ due to the deposition of the molecules. (**b**) $R_{S}$ (solid line) and $R_{C}$ (dashed line) of pristine CVD graphene before (green) and after (red) deposition of Pentacene (Chip 2). The shaded areas are the standard errors of the estimated slope and intercept of the linear regression method used to extrapolate $R_{S}$ and $R_{C}$ with the Transfer Length Method (TLM) from the datasets presented in Figure 3a–d. (**c**) Schematic of the residual p-doping of graphene due to C60 deposition (Chip 1). (**d**) Schematic of the residual n-doping of graphene due to Pentacene deposition (Chip 2).

**Table 1 nanomaterials-13-01134-t001:** Summary of the electrical properties of the GFETs before and after deposition of the C60 and Pentacene molecules. The table shows the mean and variance of the normal distributions of $V_{GS}^{Dirac}$, $\mu_{e}$ and $\mu_{h}$ in Figure 3e–j. The residual doping of graphene is calculated considering the mean and variance values of $V_{GS}^{Dirac}$ in Equation (1) before and after deposition of the molecules. The Fermi energy shift ${\Delta E}_{F}$ is calculated using the energy dispersion relation of graphene and residual doping. The table shows the minimum and maximum values of the gate-dependent sheet resistance ($R_{S}$) and contact resistance ($R_{C}$) presented in Figure 4. The contact resistivity ${\rho_{C}=R}_{C}W$ is calculated using the channel width *W* = 5 μm.

	Chip 1: C60-GFETs		Chip 2: Pentacene-GFETs	
Property (Unit)	Gr	Gr/C60	Gr	Gr/Pentacene
$V_{GS}^{Dirac}$ (V)	2.8 ± 9.1	23.0 ± 5.0	1.3 ± 6.7	−7.6 ± 7.5
$\mu_{e}$ (cm^2^V^−1^s^−1^)	1430 ± 354	713 ± 261	1101 ± 227	922 ± 226
$\mu_{h}$ (cm^2^V^−1^s^−1^)	2298 ± 399	2389 ± 333	1910 ± 310	1268 ± 275
$n_{0}$ (10^12^ cm^−2^)	0.20 ± 0.65	1.65 ± 0.36	0.09 ± 0.48	0.55 ± 0.54
ΔE (meV)	−52 ± 55	−150 ± 15	−36 ± 53	+86 ± 35
$R_{S}$ (kΩ)	0.76–2.90	0.54–3.09	0.82–2.80	1.28–2.96
$R_{C}$ (kΩ)	0.23–1.01	0.14–0.74	0.19–0.91	0.28–0.85
$\rho_{C}$ (kΩμm)	1.15–5.05	0.70–3.70	0.95–4.55	1.40–4.25

## Data Availability

The data presented in this study are available on request from the corresponding authors.

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
