# Peer review of "The Effect of C60 and Pentacene Adsorbates on the Electrical Properties of CVD Graphene on SiO_2"

_nanomaterials, 2023, doi:10.3390/nano13061134_

Round 1
Reviewer 1 Report
This paper reported effects of thermally evaporated C60 and Pentacene thin films on electrical transport properties of CVD graphene on SiO2/Si substrate. It shows that a C60 thin film would increase the graphene hole-density as electron acceptor, showing p-doped graphene transport characteristics, whereas a Pentacene thin film would increase the graphene electron-density as electron donor, showing n-doped graphene transport characteristics. The presented data seems to be reasonable, and it is a large amount of input for preparing and measuring so many GFETS. However, the reviewer suggests that it still need some revisions before publication in the journal of Nanomaterials.
(1) It is reported elsewhere that the doping of graphene would cause the decrease of graphen sheet resistance mostly. Why here the sheet resistance of graphene increased instead after doping by either C60 or Pentacene thin film. It is not convincing to attribute this only to the reduction of charge mobility.
(2) As for the morphology features of the CVD graphene/C60 deposited, the AFM shows clearly graphene features with winkles and Cu grain boundaries, as demonstrated on line 140-141. It is reasonable that graphene wrinkles can be observed through AFM. However, how to observe the Cu grain boundaries through AFM measurement on transferred graphene on SiO2 substrate.
Reviewer 2 Report
The work is well structured and contains rich statistical data. The results are clearly stated. I have just one question. How ohmic contact was provided to the gold contact pads, after deposition of C60 and Pentacene films.
Reviewer 3 Report
In this paper, the authors studied the effect of thermal evaporation of C60 (n type) and pentacene (p type) films under vacuum on the in-plane charge transport properties of large area CVD graphene. This study is done on a population of 300 graphene field effect transistors. The output characteristic of the transistors reveals that a C60 thin film adsorbate increases the graphene hole-density by (1.65 ± 0.36) x 1012 cm-2, whereas a Pentacene thin film increases the graphene electron-density by (0.55 ± 0.54) x 1012 cm-2. Hence, C60 induces a graphene Fermi energy downshift of about 100 meV, while Pentacene induces a Fermi energy upshift of about 120 meV. In both cases, the increase of charge carriers is accompanied by a reduced charge mobility, which results in a larger graphene sheet resistance of ca. 3 kΩ at the Dirac point. Interestingly, the contact resistance, which varies in the range 200 Ω – 1 kΩ, is not significantly affected by the deposition of the organic molecules. I believe that publication of the manuscript may be considered only after the following issues have been resolved.
1. In order to better highlight the advantages of this work, the author needs to provide a table to compare related work.
2. Vacuum thermal evaporation is a relatively complex physical and chemical reaction process. How did the author determine the optimal process in the sample preparation process?
3. The introduction can be improved. The articles related to some applications of graphene materials should be added such as Sensors 2022, 22, 6483; ACS Sustain. Chem. Eng. 2015, 3, 1677–1685; Diamond & Related Materials 128 (2022) 109273; Talanta 2015, 134, 435–442.
4. Please check the grammar and spelling mistakes of the whole manuscript.
Round 2
Reviewer 3 Report
Accept in present form.